# Safety and Nutritional Risks Associated with Plant-Based Meat Alternatives

**Diana Bogueva [1,2,\*] and David Julian McClements [3]**

1   Curtin University Sustainability Policy (CUSP) Institute, Curtin University, Perth 6102, Australia
2   Centre for Advanced Food Engineering (CAFE), University of Sydney, Sydney 2006, Australia
3   Department of Food Science, University of Massachusetts, Amherst, MA 01003-9313, USA
\*   Correspondence: diana.bogueva@curtin.edu.au

**Abstract:** The market for plant-based meat alternatives is growing to meet consumer demands for a more sustainable, ethical, and healthy diet, as well as to address global food security issues linked to an increasing global population and climate change. Increased consumption of plant-based meat products raises questions about potential food safety risks, including concerns about allergenicity, toxicity, foodborne pathogens, and adequate nutritional composition. From a public health perspective, there has been limited research on the nutritional and health aspects of plant-based meat products, and studies of potential food safety risks of these novel protein sources are not well documented. Much of the research on the nutrition and safety of these foods has been commissioned or funded by companies developing these products, or by other organizations promoting them. This article reviews the existing literature and analyses the potential food safety and health risks associated with plant-based meat products, including nutritional, chemical, microbiological, and allergen concerns. This review has revealed several research gaps that merit further exploration to inform the conversation around the future development and commercialization of plant-based meat substitutes. Further research, technological advancements, food standards, and risk assessment and a multidisciplinary approach are essential to address safety concerns and facilitate the responsible use of new-generation plant-based meat alternatives, particularly for emerging foods with limited knowledge of their risks and benefits.

**Keywords:** plant-based meat alternative (PBMA); nutritional risks; safety risks; ultra-processed; allergies; microbial concerns; antinutrients; mycotoxins; PBMAs processing

## 1. Introduction

Growing consumer demand for more plant-based diets is propelling advancements in the development of plant-based meat alternatives (PBMAs), which are typically formulated from plant proteins and other ingredients [1].

The food industry is developing a variety of PBMA products that are usually designed to mimic the appearance, texture, mouthfeel, and taste of real meat products. Consumers are increasingly adopting these products because of their concerns about the health [2], environmental [3], and animal welfare [4] impacts of traditional meat. The current patterns of meat consumption have been linked to undesirable environmental consequences (like greenhouse gas emissions, pollution, and biodiversity loss), as well as deleterious human health effects (like cardiovascular disease, cancer, and obesity) [5]. Transitioning to a more plant-based diet could eliminate or reduce these problems [6].

PBMAs are being marketed as a means of promoting the transition away from animal-based products, and to establish a contemporary food system that benefits humans, animals, and the environment. Embracing a plant-centric diet in the wealthiest nations, despite representing just 16 percent of the world's population, has the potential to reduce greenhouse gas emissions by approximately 61 percent, while also enhancing carbon sequestration [7].

The meat substitute segment is the highest contributor to the rise of the plant-based market. In 2020, the global consumption of alternative proteins reached approximately 13 million metric tons [8]. The PBMA product category is poised to exhibit distinctive physical, functional, nutritional, and sensory characteristics. To achieve this in a cost-effective manner on a mass scale, the food industry must adeptly discern the right ingredient combinations and manufacturing processes to match the market trends. The projected growth for the global market for plant-based foods (primarily PBMAs and beverages) indicates an estimated value of USD 162 billion by 2030, a substantial increase from the USD 4.6 billion recorded in 2018 [9] and the USD 29.4 billion registered in 2020 [10]. These significant growth figures show a strong long-term outlook for investments in alternative proteins, despite their still small sector [11] and the fact that consumer demand is currently outpacing the industry's supply chain capabilities [12].

The increasing introduction of PBMAs into the food supply chain brings potential new food safety and nutritional risks that could lead to new health problems. This is why a profound understanding of the molecular and physicochemical attributes of plant-derived ingredients is imperative. Consequently, it is important that the emerging plant-based food industry carefully craft its products and considers food safety and nutrition issues and promotes healthiness. These issues depend on the nature of the ingredients and processes used to create PBMAs, necessitating control over their nutrient composition, digestibility, and bioaccessibility [1] and must, therefore, be considered on a case-by-case basis [6]. Researchers have highlighted the importance of providing consumers with information about the nutritional quality of PMBAs when compared to real meat, so they can make more informed decisions [13,14].

This review aims to critically evaluate the potential safety and nutritional risks associated with the production and consumption of PBMA products. It highlights potential safety and nutritional risks associated with the main production stages of these products: (1) protein isolation and functionalization; (2) product formulation; (3) processing; and (4) storage.

## 2. Methodology

A systematic literature review based on scientific articles published between January 2018 and May 2023 was carried out. Two databases, Scopus and Web of Science, were searched for articles related to food safety risks and nutritional aspects of PBMAs. The review involved a careful analytical process selecting the concept grid and list of keywords to be explored, formulating the research inclusion and exclusion criteria for articles' selection and elimination, and searching databases (Table 1). The study selection process was managed using the Covidence platform [15]. All the records obtained from the database searches using Scopus and Web of Science were uploaded to Covidence. All the duplicated records were automatically removed. According to the set eligibility criteria, the title and abstracts were screened first and then, if selected, the full texts were screened. The review aimed to (a) identify all previously published work on the safety and nutritional risks linked with plant-based meat alternatives; (b) determine the consensus and controversies related to these issues; (c) aggregate empirical findings to support evidence-based issues.

Articles were evaluated based on their final content, which resulted in further exclusion. A total of 326 articles were available from the search of the databases, and after the removal of duplicates and screening of the titles and abstracts, 48 articles were selected for inclusion. These articles all dealt with potential safety and nutrition risks associated with plant-based meat alternatives.

**Table 1.** Inclusion and exclusion criteria used during the literature search.

| Search Criteria |
| --- |
| Search engines:<br>Scopus<br>Web of Science |
| Keywords:<br>("plant-based meat") AND ("safety risk" OR "allergen" OR "chemical" OR "microbiological" OR "health risk" OR "nutritional risk) |
| Search type:<br>All fields<br>Article title, abstract, and keywords |
| Timeframe:<br>January 2018–May 2023 |
| Language:<br>Full-text downloadable papers written in English |
| Study focus:<br>Focus on the safety and nutritional risks of plant-based meat |
| Inclusion criteria:<br>Studies published in peer-reviewed journals<br>Empirical studies or reviews with strong relevance |
| Exclusion criteria:<br>No discussion of plant-based meat safety and nutrition-related aspects<br>Conference papers, abstracts, and educational papers<br>Studies with a different focus |

The search screened categorically and descriptively the available literature to collect information and finalize the sample for the study (*n* = 48). Table 2 below outlines the selected articles. The selected articles focused on the nutritional risks (*n* = 27) and the safety risks (*n* = 6) of PBMAs, with some of them sharing information and analysis on both risks (*n* = 15). The published studies were from around the world, including Australia, USA, France, Italy, Spain, Brazil, and Sweden.

**Table 2.** Summary characteristics of the included articles related to PBMAs' nutrition risks aspects discussed.

| N | Study | Reason for Selection Based on PBMA Nutrition Risks |
| --- | --- | --- |
| 1 | Ogawa et al., 2018 [16] | Discuss that understanding the mechanism of the gastrointestinal fate of PBMAs is of paramount importance to gain a better understanding of their digestibility and bioavailability. |
| 2 | Curtain and Grafenauer, 2019 [17] | Analyze 137 PBMAs (61% Australian made) and found these products to be higher in carbohydrates, sugars, and sodium. PBMAs lack equivalence with similar meat products and fall short of key nutrients. Of consideration are PBMAs fortification and the need for industry guidance and nutritional regulations (e.g., iron, B12). |
| 3 | McClements, 2020 [18] | Some PBMAs contain more calories, total fat, saturated fat, and salt, as well as less protein, essential amino acids, vitamins, and minerals than real meat. PBMAs can also contain more carbohydrates, sugar, and salt. |
| 4 | Van Vliet et al., 2020 [19] | Discuss the multitude of unlisted compounds in PMBA ingredients and their potential impact on human health, including the absence of nutrients, like creatine. Adequate intake of zinc, copper, and vitamins A and D from natural foods is linked to reduced cardiovascular disease and overall mortality risk, but these nutrients in PMBAs may not offer the same benefits. Supplements of carotenoids, vitamin A, and calcium in PMBAs may not decrease the risk of cancer or heart disease and could even raise it. This suggests that consuming nutrients outside their natural food sources may not be the best approach for promoting health. |

**Table 2.** *Cont.*

| N | Study | Reason for Selection Based on PBMA Nutrition Risks |
|---|---|---|
| 5 | Lee et al., 2020 [20] | Digestibility and gastrointestinal fate of plant-based meat analogs need further investigation. |
| 6 | Takefuji, 2021 [21] | Argue plant-based meat has comparable calorie and saturated fat levels to conventional meat, with higher sodium content that can raise the risk of high blood pressure when consumed excessively. Moreover, manufacturers often use various additives to mimic the texture and flavor of traditional meat. This complexity highlights the need for plant-based meat producers to be transparent about the health aspects of their products and communicate them effectively to consumers. |
| 7 | Harnack et al., 2021 [22] (Based on USA market) | Assess the nutritional quality of 37 plant-based ground beef alternatives in the United States and compare their leanness to conventional ground beef. Deliberate on the high sources of dietary fiber, iron, manganese, copper, folate, and niacin while maintaining a low saturated fat content, but they tend to be high in sodium and generally contain less protein, zinc, and vitamin B12 compared to traditional animal-source beef. |
| 8 | Lacy-Nichols et al., 2021 [23] (Based on Australian market) | Examine claims related to nutrients, ingredients, processing, and health. Most claims revolved around meat-related nutrients, with 94% featuring protein claims and 30% having cholesterol claims. Additionally, 74% touted being GMO-free, and 63% promoted a plant-based identity. Some companies elaborated on these claims and discussed ingredient health benefits on their websites. Acknowledging and explaining the presence of ultra-processed ingredients and additives is crucial for manufacturers. |
| 9 | Alessandrini et al., 2021 [24] | Discuss many plant-based meat alternatives are higher in sodium and saturated fat compared to meat. |
| 10 | Toribio-Mateas et al., 2021 [25] | In a small randomized controlled trial involving 20 participants, it was found that not all PBMA products are necessarily highly processed and harmful to the human gut microbiome. Occasional substitution of animal meats with PBMA in flexitarian diets may benefit the gut microbiome. However, more rigorous and well-defined research is required. |
| 11 | Cole et al., 2022 [26] | Present a comprehensive analysis of 117 U.S. burger products, including 28 plant-based and 89 veggie burgers, conducted to compare their nutritional profiles with traditional burgers. Veggie burgers featured a wider range of ingredients, had higher carbohydrate levels, increased sodium and sugar content (less favorable), and unusually high vitamin C content due to the inclusion of peppers and rosemary extract. |
| 12 | Toh et al., 2022 [27] | Discuss the limited nutritional information on PBMAs products, the many different ingredients, and a higher degree of processing. Limited information on contents of iron and vitamin B12, and whether preservatives are present. Need for identification of the long-term health effects of consuming PBMAs. |
| 13 | Ishaq et al., 2022 [28] | Discuss the gastrointestinal fate of meat analogs not being exhaustively investigated, and this knowledge gap hinders our ability to gain a comprehensive understanding of the nutrient bioavailability associated with these products. |
| 14 | Bryngelsson et al., 2022 [29] (Based on Swedish market) | Assessed 142 PBMAs and argue many lack micronutrient information and significant variation exists among product categories. The bioavailability of iron may also be hampered by the presence of antinutrients, e.g., phytic acid available in soy concentrates, and requires thorough assessment. Concerns arise about lower methionine levels in PBMAs, an essential amino acid found in regular food. Lowering salt content in PBMAs is desirable, but could impact taste due to the off-flavors in plant proteins necessitating masking. Analyzing fiber type (soluble/insoluble) and exploring fortification and processing are also advised for enhancing PBMAs. |
| 15 | Zhao et al., 2022 [30] | Discuss the need for extensive research into the nutritional composition and processing attributes of lipids and polysaccharides, particularly in the context of optimizing their utilization as fat substitutes and binders. This entails investigating more readily available sources of lipids and polysaccharides, fine-tuning the nutritional ratios, functional characteristics, and targeted applications of these fat substitutes. This direction is crucial for advancing the future development of PBMAs. |
| 16 | Harnack et al., 2022 [31] | Discuss that PBMAs in general contain less protein and of lower quality, mainly lower iron bioavailability and less zinc, vitamin B12 (fortified), and potassium than animal meats and sodium in high amounts. Dietary guidance around the place of PBMAs is needed. |

**Table 2.** *Cont.*

| N | Study | Reason for Selection Based on PBMA Nutrition Risks |
|---|---|---|
| 17 | Penna Franca et al., 2022 [32] (Brazilian based) | Discuss saturated fat (to mimic the marble effect of meat), protein, additives, and sodium content are higher for second-generation PBMAs products. Argue the forthcoming iterations of meat substitutes should prioritize two critical aspects: lowering saturated fat levels and minimizing the use of additives. The processing involved in creating these substitutes may lead to a reduction in dietary fiber content within protein ingredients sourced from soy and pulses. According to NOVA classification, textured and isolated proteins, along with gluten, fall under the category of industrial formulations, meaning all 1st- and 2nd-generation PBMAs are classified as ultra-processed food. |
| 18 | Cutroneo et al., 2022 [33] (Italian based) | Discuss the PBMAs longer list of ingredients and significantly higher salt content. Comment on the need for improvement of the formulation of meat analogues in terms of the number of ingredients added and processing. |
| 19 | Zhou et al., 2023 [34] | Argue PBMAs have not been designed to improve human health and well-being and may have a dietary inadequate nutritional profile, cause problems with digestibility and absorption rates, and pose unforeseen nutritional consequences. Uses the INFOGEST method to monitor the digestion and/or bioavailability of proteins, lipids, carbohydrates, vitamins, and minerals of food including PBMAs, but note that the method needs improvement to be more reliable and accurate. |
| 20 | Lawrence et al., 2023 [35] (Australia based) | Comment that transitioning to plant-based 'milk' and 'meat' alternatives may have unintended consequences on nutrient intake, potentially raising the risk of nutritional deficiencies within the Australian population. It underscores the significance of promoting a diverse array of food sources to fulfill nutrient requirements when adopting a plant-based diet, as opposed to merely substituting animal-derived meat and dairy products with visually similar plant-based alternatives. |
| 21 | Flint et al., 2023 [36] | Discuss the ultra-processed nature of PBMAs and express concerns around the potential problems with PBMAs regarding their digestibility and suitability for children (particularly regarding nutritional needs) and a lack of clarity in relation to their health value. Express concerns related to higher levels of saturated fat, salt, and free sugar content and inclusion of additives, such as artificial colors, flavors, and preservatives, for human health. Conclude PBMAs health value needs further consideration. |
| 22 | Rizzolo-Brime et al., 2023 [37] (Spain market focused) | Review 148 products, noting that the majority of them exhibit low sugar content, but moderate levels of carbohydrates, total fat, and saturated fat (e.g., coconut oil or palm oil). These products also tend to contain high quantities of sodium, oil, and various additives, including coloring, flavoring, and binding agents. Note that PBMAs generally feature a lengthy list of ingredients and additives, placing them in the category of ultra-processed foods (UPFs), according to the NOVA classification system. Further research is required to determine whether these UPFs could serve as a viable option for healthier and more sustainable dietary patterns compared to traditional meat consumption. Establishing a regulatory framework for PBMAs is crucial to empower consumers with better food choices. |
| 23 | Melville et al., 2023 [38] (Australia based) | Study 132 PBMAs and revealed that meat analogues also fall into the category of ultra-processed foods, with only a limited number fortified with essential micronutrients typically found in meat. Additional research is required to comprehensively assess the health implications of consuming these products. |
| 24 | Romão et al., 2023 [39] | Reviewed 11 studies on food label accuracy and found worldwide food label regulations allow discrepancies in nutritional values. Meat substitutes contain higher carbohydrates than meat, but excessive fiber can harm taste and commercial appeal. Dietary fiber's hygroscopicity affects cooking oil retention, potentially increasing fat content during grilling or frying. PBMAs have similar total and saturated fat levels as animal-based products, contributing to sensory qualities and shelf life. Fat-free meat substitutes are impractical due to taste and desirability concerns. Excessive sodium in meat substitutes contributes to high sodium intake, necessitating sodium-free alternatives. There is a need for more research on global meat substitute nutritional composition. |
| 25 | Salomé et al., 2023 [40] (French based) | Discuss PBMAs can be levers for healthy diets only when well nutritionally designed with enough zinc and iron for a substantial red meat reduction. Choosing the correct ingredients can result in a nutritionally highly effective meat substitute. |

**Table 2.** *Cont.*

| N | Study | Reason for Selection Based on PBMA Nutrition Risks |
|---|---|---|
| 26 | McClements and McClements, 2023 [41] | Discuss PBMAs highly processed nature containing elevated levels of saturated fat, sugar, starch, and salt, alongside lower levels of micronutrients, nutraceuticals, and dietary fibers. This composition can lead to the rapid digestion and absorption of macronutrients, like starch and lipids, potentially resulting in disruptions to hormonal and metabolic systems. It becomes crucial to design the food matrix of PBMAs with nutrient profiles that match or surpass those of animal-based foods. Additionally, fortifying these alternatives with dietary fibers and nutraceuticals can further enhance their nutritional value, digestibility, and bioavailability. |
| 27 | Rizzo et al., 2023 [42] (Italian based) | Discuss the lack of understanding regarding how health-related factors influence consumers' choices when it comes to PBMAs. It is imperative to consider the possibility of organic certification for PBMA products, particularly because consumers often raise questions about the healthiness of highly processed organic food products. |

As outlined in Table 3 below, there were only six (*n* = 6) studies identified focused on the PBMAs safety risks associated with a variety of aspects, such as microbial proliferation, pathogenic bacteria, mycotoxins, and allergies. These aspects are discussed later in the paper in more detail.

**Table 3.** Summary characteristics of the included articles related to PBMAs' safety risks aspects discussed.

| N | Study | Reason for Selection Based on PBMA Safety Risks |
|---|---|---|
| 1 | Tóth et al., 2021 [43] | Discuss the less explored food safety aspects of PBMAs of faster microbial proliferation in hot meals with meat analogues, posing a slightly higher risk compared to meat. Microbial growth in cooked PBMAs was observed after 12 h, with yeast and Enterobacteriaceae exclusively found in meatless options, indicating a somewhat elevated food safety concern compared to meat-containing dishes. Quantifying yeasts, molds, and Enterobacteriaceae can be valuable for monitoring hygiene during meat analogue production. Enhanced attention is required for the preparation, processing, and storage of PBMAs to ensure their safety. |
| 2 | Hadi and Brightwell, 2021 [44] | Discuss that plant-based meat can potentially harbor pathogenic bacteria originating from the raw ingredients used in PBMAs. Note that some endospore-forming bacteria, such as *Clostridium* spp. or *Bacillus* spp., may withstand the extrusion process' heating, posing health risks. Additionally, anti-nutrients, like protease inhibitors, α-amylase inhibitors, lectins (phytohemagglutinin), polyphenols (particularly tannins), and phytic acid, found in PBMAs can impact gut function, affecting digestive enzymes, iron absorption, and endocrine systems. Legumes, a common component of PBMAs, can also carry allergens, leading to mild-to-severe reactions affecting the skin, gastrointestinal, cardiovascular, and respiratory systems. It remains uncertain whether thermal processing can reduce the allergenicity of legume proteins, such as those from soybeans and peas, necessitating further clinical studies for clarification. |
| 3 | Mihalache et al. 2022a [45] (Italian study and EFSA based data) | Discuss the necessity of establishing a comprehensive regulatory framework for mycotoxins in PBMAs and stress the importance of updating regulations and conducting thorough risk assessments for natural toxins in PBMAs. Note that consumption of contaminated with mycotoxins soy-based alternatives could lead to development of liver cancer. Further studies on natural toxins exposure related to other types of PBMAs (e.g., peas, chickpeas) is needed. |
| 4 | Mihalache et al., 2022b [46] | Discuss PBMAs may carry chemical contaminants and natural toxins with potential adverse effects. Unfortunately, there is insufficient research on contamination in PBMAs. Instances of mycotoxin (e.g., soy-based products) and alkaloid (e.g., tropane and β-carboline) contamination have been observed. Future research is crucial for gathering data on natural toxin contamination in PBMAs and exploring regulatory measures, like those in the EU. |

**Table 3.** *Cont.*

| N | Study | Reason for Selection Based on PBMA Safety Risks |
|---|---|---|
| 5 | Kopko et al., 2022 [47] | Highlight that PBMAs have the potential to introduce new allergens to both allergic and non-allergic consumers. Emphasize the risk associated with (new) sensitization and allergies to novel proteins that may emerge during product formulation. Mention the challenge regarding ingredient identification, as ingredients present at less than 2% in finished products do not need to be disclosed if the source is not recognized as a major allergen. It requires a science-based allergenicity assessment of new foods to protect consumers and the need for an internationally established thresholds reference doses. More work is required in addressing known allergens arising from novel protein sources. |
| 6 | Liu et al., 2023 [48] | Describe PBMAs as not sterile and mention possibility of potential microorganism's introduction through the addition of other raw ingredients (e.g., vitamins, minerals, flavoring, colors) or through post-processing contamination. Discuss PBMAs support of the growth of spoilage and pathogenic bacteria and foodborne pathogens, such as *E. coli*, salmonella, etc. Pathogens grew better in pea-based meat analogues than soy-based analogues. PBMAs, regardless of the type of plant protein, contain lower indigenous microbial loads compared with meat. |

Some of the articles picked out (*n* = 15) discussed issues related to both nutritional and safety issues associated with PBMAs. These are outlined in Table 4 below. Among the discussed risks are nutrients deficiency, allergens, sodium level, additives, digestive issues, and so on. These topics will be covered in greater detail later.

**Table 4.** Summary characteristics of the included articles related to both PBMAs' safety and nutrition risks aspects discussed.

| N | Study | Reason for Selection Based on PBMA Nutrition and Safety Risks | |
|---|---|---|---|
| 1 | Fresán et al., 2019 [49] | Comment on the diverse allergic potential of PBMAs, including the necessity for some individuals to avoid wheat-based PBMAs due to medical conditions, like celiac disease, non-celiac gluten sensitivity, and wheat allergy or specific dietary preferences. such as gluten- and grain-free diets. Additionally, discuss the avoidance of soy- and nut-based meat analogs, highlighting the common use of GMO soybeans in commercial products and the preferences of consumers who follow a non-GMO diet. | The nut-based analogs were higher in total fat, monounsaturated fat, and niacin. Additionally, PBMAs approach the daily sodium intake limit. |
| 2 | Bohrer, B. M. 2019 [50] | Examines the challenges in assessing potential health benefits of carbohydrate ingredients, like methylcellulose, gum acacia, xanthan gum, and carrageenan, in meat analog products. It concludes that despite challenging the health and wellness of these products, no significant health risks or concerns have been identified. Minimally processed soy protein typically darkens the color of meat products and can elicit a bitter flavor. | Discuss PBMA ingredients, which fall under the category of ultra-processed foods with lengthy ingredient lists. Besides added dietary fiber, there are no significant nutritional advantages over traditional beef burgers on a macronutrient level. PBMAs contain high levels of non-dietary fiber carbohydrates. Highlight concerns about the growing consumption of ultra-processed foods and reduced intake of whole foods, which may lead to unforeseen nutritional consequences. Conclude that modern meat analogs provide a similar nutrient composition to traditional meats, albeit with numerous ingredients and extensive processing. |

**Table 4.** *Cont.*

| N | Study | Reason for Selection Based on PBMA Nutrition and Safety Risks | |
|---|---|---|---|
| 3 | He et al., 2020 [14] | Address the insufficient scientific data on PBMA safety, stressing the importance of further investigation and chemical safety assurance. Note that PBMA's high moisture and neutral pH create conditions favorable for microbial growth, especially at high temperatures. Contamination risks exist from the environment and nonsterile ingredients after extrusion. Suggest adopting storage and handling practices for PBMAs akin to those for regular meat. | Discuss the need of additional scientific evidence supporting the health properties of PBMA, identifying more appropriate protein sources to enhance product quality, refining appearance and flavor, investigating structure formation mechanisms during extraction or shearing processes, and establishing methods and standards for evaluating PBMA quality. |
| 4 | Luchansky et al., 2020 [51] | Examine how PBMA storage at 4 °C may promote the growth of foodborne pathogens, such as *E. coli* O157:H7, *Salmonella* spp., and *L. monocytogenes*, due to lower background microorganism levels. Note that plant-based burgers tend to have better pathogen viability during cold storage than beef-based ones. Emphasize the need for additional research and data to understand how PBMA product formulation impacts the presence, levels, and types of native and harmful microbes. This is essential for enhancing safety, quality, and shelf life. | Discuss that the values for pH, salt, carbohydrate, and nitrite were notably higher for plant burgers. |
| 5 | Santo et al., 2020 [52] | Highlight potential issues associated with process-induced hazardous chemicals. | Recognize that the support for the benefits of meat substitutes may not be as robust as some claims suggest, emphasizing the need for further research in this area. |
| 6 | Rubio et al., 2020 [53] | Express concerns about including LegH in PBM due to correlations between heme iron intake and a heightened risk of diabetes. | Explore factors within plant-based proteins that have been identified as potentially reducing nutrient bioavailability after consumption. These factors encompass structures that resist proteolysis, protein conformation, and antinutrients, such as tannins, phytates, and lectins. |
| 7 | Tso and Forde, 2021 [54] | Present possible problems with process-induced hazardous chemicals | Make the case that diets centered around novel plant-based substitutes tend to fall short of daily requirements for essential nutrients, like calcium, potassium, magnesium, zinc, and vitamin B12, while surpassing the reference diet in terms of saturated fat, sodium, and sugar content. Express concerns about the necessity of gaining a better understanding of how PBMAs behave within the human gastrointestinal tract to address these nutritional challenges. |
| 8 | Sun et al., 2021 [55] | Discuss the antinutritional factors and allergenic potential of soy proteins, and also wheat allergy, one of the most common causes of food allergies in children. | Discuss the plant proteins lack of one or more of the essential amino acids. |

| N | Study | Reason for Selection Based on PBMA Nutrition and Safety Risks |
|---|---|---|
| 9 | McClements and Grossmann, 2021 [1] | Examine the potential toxicity associated with fortifying foods with unsaturated fatty acids, as this can render them susceptible to lipid oxidation and the formation of rancid byproducts during storage and processing. Emphasize the need for further research to understand how differences in protein digestion, absorption, amino acid profiles, and allergenicity impact human health and overall well-being. | Discuss the unique molecular, chemical, and physical properties of plant-derived ingredients diverge significantly from their animal-derived counterparts. Understanding these distinctions, including characteristics like off-flavors, unpleasant mouthfeels, poor solubility, and inconsistent performance, is crucial. Many plant-based fats are naturally high in unsaturated fatty acids, remaining liquid at room temperature. Attempting to solidify them through hydrogenation processes is undesirable, as it can lead to the formation of trans- or saturated fatty acids, linked to an increased heart disease risk. Consequently, some producers turn to coconut oil, which contains substantial levels of saturated fats, potentially posing health concerns. Additional research is needed to unravel how specific protein blends behave in the human gut when integrated into various plant-based food matrices. Their nutritional effects depend on amino acid profiles and how they digest in the upper gastrointestinal tract. The scarcity of high-quality plant-based ingredients with the necessary functional attributes poses a challenge. Discrepancies in nutrient compositions among different PBMAs could carry health implications when consumed as part of a long-term diet. |
| 10 | Tyndall et al. 2022 [56] | Discuss the need to investigate the extended shelf life of PBMAs, focusing on mitigating deteriorative reactions that impact technological properties. Additionally, conducing toxicological tests is essential to guarantee the safety of new and innovative ingredients. | Discuss PBMAs inadequate nutrients and low calcium, potassium, magnesium, zinc and vitamin B12, while containing higher levels of sodium and saturated fat compared to meat. Raise the question for the need for identification of the long-term health effects of consuming PBMAs including the long list of ingredients, such as macronutrients (proteins, fats, carbohydrates), micronutrients, flavoring agents, emulsifiers, colors, salt, and plant-based extracts. Plant-based proteins are less digestible and lower in certain essential amino acids. The choice of fat type also impacts nutritional quality, with saturated and harder fats being less desirable. Further research is needed to understand the long-term health effects of PBMAs. |
| 11 | D'Alessandro et al., 2022 [57] | Access on average, each processed plant-based product contains approximately 1.84 additives. Note that products advertised as "healthy" and "organic" often undergo chemical manipulations to facilitate preservation and enhance appearance. The most commonly utilized additives in these products are stabilizers, thickeners, and emulsifiers. | Review 560 PBMAs products. On average, a lower protein and higher fat, salt, and carbohydrate content, and an almost constant presence of additives. Fat content in plant-based burgers, meatballs, and cutlets is more than double, with a prevalence, however, of unsaturated fatty acids. Concerns raised about the methodologies applied to improve PBMAs conservation and palatability. |

**Table 4.** *Cont.*

| N | Study | Reason for Selection Based on PBMA Nutrition and Safety Risks | |
|---|---|---|---|
| 12 | Nezlek and Forestell, 2022 [58] | Mention studies claiming possible problems with process-induced hazardous chemicals. | Discuss the PBMA highly processed nature, including high saturated fats, and inability to offer the same nutritional benefits of the foods from which they are derived (e.g., legumes and soybeans). PBMA may not contain the same amounts of quality protein as meat and are not true nutritional replacements for meat. Comment that researchers need to be cautious when drawing conclusions about meat substitutes. |
| 13 | Shaghaghian et al., 2022 [59] | Discuss that the consumption of gluten-containing PBMAs products carries a potential health risk for individuals with gluten sensitivity, intolerance, or allergies. Formulations with soy proteins also can lead to allergies. Using legumes as functional ingredients may promote gastrointestinal discomfort due to containing substances, e.g., raffinose. The selection of the most suitable plant protein source main production process is crucial for creating PBMAs with the best bioavailability, digestibility, and functional qualities. | Discuss lack of information about the digestibility and bioavailability of plant-based proteins intended for use in PBMAs. Amino acids bioavailability in some sources of plant proteins are limited. Careful selection of appropriate protein sources and processing technology to improve the digestibility and bioavailability of plant-based proteins is needed for overall PBMA improvement. Consideration of antinutritional compounds in plant-based proteins is also required. Comment the need to develop effective strategies for improving the nutritional value, digestibility, bioavailability, and functional properties of plant proteins from both conventional and emerging sources. |
| 14 | Ahmad et al., 2022 [60] | Discuss toxicants and poisons (e.g., heterocyclic aromatic amines) produced during the high-temperature processing of protein-rich foods, and PBMAs vulnerability to harmful chemicals because of their high protein content. Note the presence of some allergic plant proteins (e.g., soybean protein) can cause a from light-to-severe, depending on the degree of sensitivity, health hazard. Same is reported for people with gluten sensitivity, intolerance, and allergy as potential threat aspects for coeliac disease. PBMAs have high-moisture content; therefore, proper storage, packaging, and microbiological safety should be investigated. | Discuss efficiency of nutrients in meat alternatives falls short compared to actual meat. Argue that incorporating numerous additives to mimic meat's texture, juiciness, mouthfeel, and flavor raises concerns about nutrition, food safety, clean labeling, costs, and consumer confidence. PBMAs may lose some nutrients due to intense processing methods, like blending, homogenization, and high-temperature cooking. They often contain excessive salt, numerous ingredients, and additives, along with undesirable off-flavors. Additionally, it is crucial to scrutinize soy-based components for anti-nutritional factors. |
| 15 | Andreani et al., 2023 [61] | Discuss several aspects of chemicals and additives requiring attention, including: Binding agents—(e.g., oleogels, starches, hydrocolloids, or fibers) used as fat replacers; Additives—mentioned the need for using many additives to mimic meat characteristics; Ingredients list—highly complex products with a long list of unfamiliar ingredients conveying the message of highly processed and unhealthy. | Discuss several aspects: Taste—could be affected by lipid oxidation and cause undesirable characteristics; Biomarkers on inflammation—no improvements of the biomarkers were identified with PBMAs consumption; Absorption and bioavailability of iron—higher iron content compared to meat; Lower protein digestibility compared to meat. |

Figure 1 uses a PRISMA flow chart to document the selection criteria used in this study. The chart maps out the number of recorded studies and visually illustrates the flow of the studies selections though each phase of the review process in sequential order. From a total number of $n = 327$ articles initially identified, the final list of articles and topics

related to safety and nutritional risks were reduced to *n* = 48 articles. These are discussed in association with the main PBMAs production stages, such as (1) protein isolation and functionalization; (2) product formulation; (3) processing; and (4) storage.

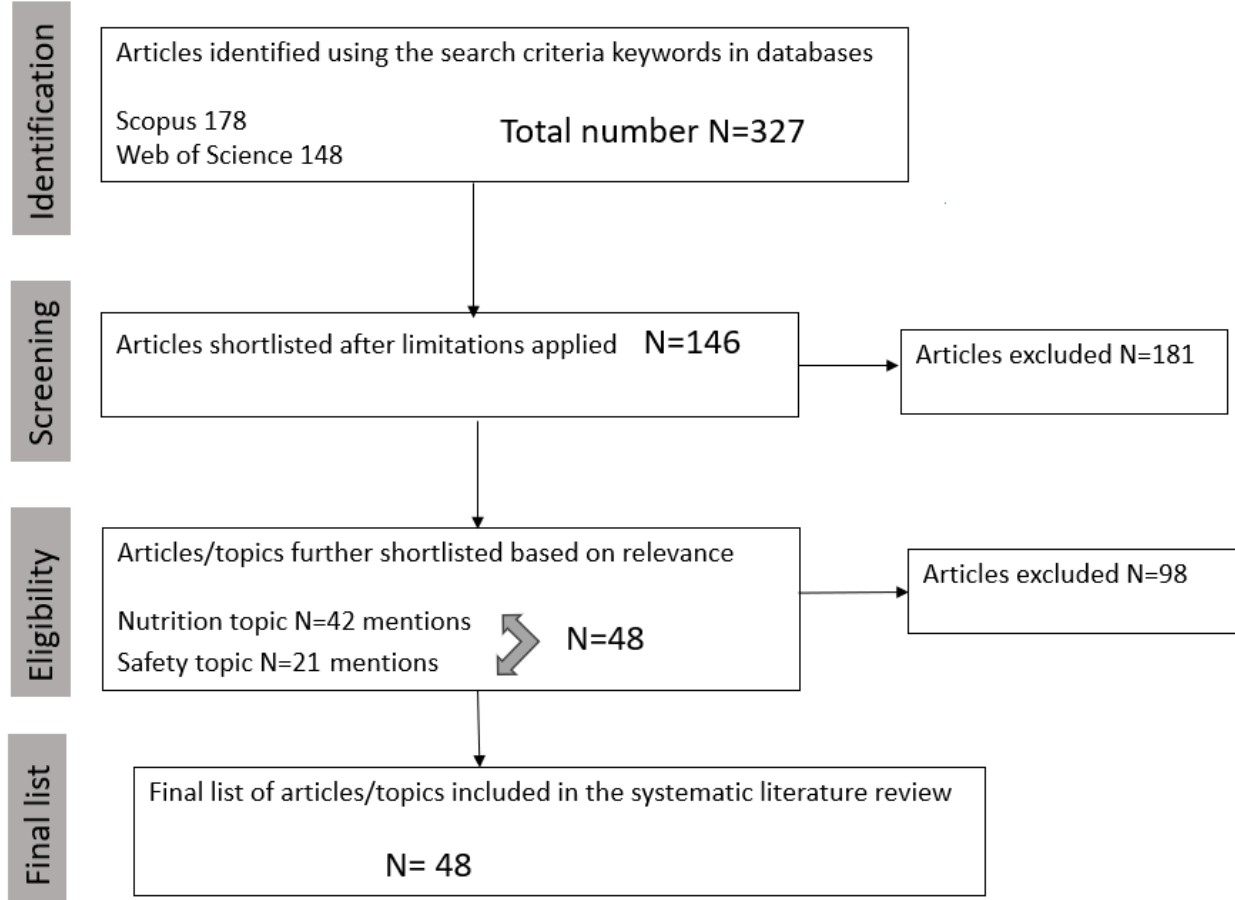

**Figure 1.** Prisma flow graph of the procedures used to select papers on the safety and nutrition of plant-based meat alternatives included in this review.

## 3. Results and Discussion

The new PBMAs are meant to be serving as substitutes for traditional meat products. Their nutritional profile, health implications, and safety risks are contingent on the specific ingredients they contain and the type of processing operations used to manufacture them [36]. The fact that they possess distinct compositions and structures compared to real meat products means that PBMAs are also probable to yield divergent effects on nutrition, health, and safety impacts.

### 3.1. How Are Plant-Based Meats Made

While there has been a significant surge in awareness of plant-based meat alternatives in recent years, it is important to note that these products have been in production and consumption for well over a millennium [62]. In antiquity, Asian civilizations pioneered various meat analogues, crafting relatively straightforward derivatives from soybeans (like tofu and tempeh) or wheat (like seitan) [53,63]. Importantly, these early versions were not specifically intended and designed as direct replacements for meat [58]. Several decades ago, texturized vegetable protein (TVP) emerged as a meat substitute, created through the extrusion of defatted soy meal, soy protein concentrates, or wheat gluten [64].

In recent times, a fresh wave of new-generation plant-based meat alternatives have emerged, primarily targeted at omnivores who seek the appealing sensory attributes of meat. Currently, PBMA products are mainly available in the form of burger patties, mince,

sausages, and nuggets, although there is a growing interest in developing more intricate and complex products, like chicken breast, beefsteak, or pork chop analogs [65]. PBMAs are typically engineered to be environmentally sustainable, while faithfully replicating the appealing appearance, feel, and taste of authentic meat products [34]. While some of them are formulated to closely mimic the nutritional composition of real meat, others may not prioritize this aspect. Moreover, most PBMAs fall into the category of ultra-processed foods, which means they may be digested and absorbed at a different rate compared to real meat. This can potentially lead to alterations in their nutritional and health effects [66]. As the popularity of plant-based diets increases, concerns and questions emerge regarding the potential safety and nutritional risks associated with transitioning from an omnivorous diet to one that is more plant based. As mentioned earlier, safety and nutritional issues can stem from the ingredients and processing operations methods employed in the production of PBMAs products [67]. Numerous potential challenges exist in the realms of food safety and nutrition when it comes to plant-based food. These challenges encompass various aspects, including the presence of different types of chemical and microbial contaminants present in the ingredients used, concerns about food adulteration issues, elevated levels of food additives, the use of genetically modified ingredients components, mislabeling, the introduction of new sources of allergens, potential vitamin or mineral deficiencies, and changes in macronutrient composition (such as protein, carbohydrate, or fat content). It is, therefore, of outmost importance to consider these issues when developing the next generation of plant-based foods. This ensures that they not only align with environmental sustainability goals, but are safe and nutritious, and beneficial for consumers and the environment.

The production of PBMAs consists of four main stages [68]:

- **Protein isolation and functionalization:** Plant proteins are extracted and purified to produce flours, concentrates, or isolates. In some cases, they undergo additional processing methods, like hydrolysis, conjugation, or denaturation, to enhance their functionality.

- **Product formulation:** The plant proteins are blended with a variety of other components, including carbohydrates, lipids, salts, flavors, and colors, to craft plant-based meat analogues that simulate the look, feel, texture, taste, flavor and functionality of actual meat products. Moreover, nutrients, like vitamins, minerals, dietary fibers, or nutraceuticals, may be incorporated to align with or surpass the nutritional composition of real meat.

- **Processing:** The mixture of ingredients goes through a sequence of processing operations that promote the development of meat-like structures and properties. These operations often involve mixing, extrusion, shearing, molding, and cutting.

- **Storage:** The product, packaging materials, and environmental conditions must be carefully designed to ensure the safety and quality of PBMAs throughout storage, transportation, and distribution. Achieving this involves careful management and control of microbiological, chemical, and physical deterioration processes.

The ultimate goal of this process is to produce a final product that faithfully replicates the desirable quality and sensory characteristics of real meat, while also ensuring its safety and nutritional value. Currently, extrusion technologies are the most widely used method of creating PBMAs that mimic the properties of actual meat. However, ongoing developments include other alternative technologies, such as shear cells, 3D printing, recombinant proteins, and mycelium [53,69]. In the following sections, we consider some of the potential health and safety risks associated with the main production stages of PBMAs. Certain safety considerations regarding novel protein sources are inherent to the product itself, but numerous potential risks can also arise from the methods of production and the conditions under which they are processed [70].

### 3.2. Protein Isolation and Functionalization

A process for isolating protein can be perceived as a series of interconnected stages, where the protein's purity increases progressively through each step. These include: (a) sourcing suitable protein and obtaining the material; (b) extracting from the source; (c) segregating from non-protein elements, like nucleic acids and lipids; (d) concentrating the primary protein and segmenting it into fractions with distinct proteins; and (e) culminating in refined methods, like chromatography, that yield the ultimate end product [71].

Proteins play a pivotal role in shaping the technological, physicochemical, and sensory characteristics of plant-based food products. For instance, they have a profound impact on attributes, such as structure, texture, appearance, chewiness, water retention, and nutritional qualities, of PBMAs [72,73]. The choice of an appropriate plant protein is crucial, as it can impart a range of desirable functional attributes to the final product, including thickening, gelling, emulsifying, structuring, foaming, and fluids retention [1].

Plant proteins are typically derived from sources like soybeans, peas, or corn to produce flours, concentrates, or isolates. Extraction and purification can be accomplished through a diverse array of techniques. There methods encompass conventional approaches utilizing aggressive chemicals, like acids, bases, and/or organic solvents; eco-friendly green extraction methods involving single or mixed enzymes; and cutting-edge physical extraction techniques, such as ultrasound-, pulse electric field-, microwave-, and high pressure-assisted extraction [74,75]. Notably, it is important to highlight that a significant number of the techniques employed for proteins extraction were not originally intended to optimize and enhance their functionality. Instead, these methods were optimized for the extraction of oil or starch from plant materials. Consequently, the functionality of proteins may be compromised, as they can become denatured or aggregated during the isolation process.

### 3.3. Product Formulations

Product formulation entails understanding how materials interact to produce enhanced properties, optimize processing efficiency, and deliver active ingredients effectively. In the context of food, formulation pertains to the art of crafting, planning, or evolving food items, with the aim of incorporating specific functionalities. These functionalities can span from conferring extra nutritional advantages to enriching food products. PBMAs are distinctly defined as products meticulously crafted to emulate the color, flavor, taste, aroma, consistency, texture, and visual characteristics of animal-derived products to match or at least closely align with the sensory encounter of consuming meat products. PBMAs are formulated to simulate the physicochemical and organoleptic characteristics of traditional meat products.

During this stage of production, plant proteins are combined with an array of other functional ingredients to achieve the desired appearance, feel, texture, and taste of the end product. These additional components include flavorings, colorings, emulsifiers, texture modifiers, gelling agents, and binding agents [50]. Due to the diverse formulation of plant-based meat substitutes involving elements like proteins, water, fats, carbohydrates, flavor constituents, binding agents, and colorants, their operational characteristics assume a crucial role in dictating their attributes [76]. Some consumers have voiced concerns regarding the extensive use of additives in PMBAs, classifying them as ultra-processed foods [66,77]. A number of the components present in these intricately formulated food products have raised nutritional concerns, including saturated fats, highly refined flours, and high salt content [41,50]. Fortification practices vary between products, and so each product should be considered on a case-by-case basis.

Concerns have also been raised by some researchers regarding the inclusion of leghemoglobin in certain PBMAs. Leghemoglobin is an iron-containing hemeprotein that can be derived from soybean root nodules and is used to impart the desirable red color and meaty flavors typically associated with the hemoglobin in real meat [2]. Given the challenges in obtaining sufficient quantities from soybeans, this protein is often produced through

genetically engineered yeast cultures. Some researchers have drawn connections between higher heme iron intake, elevated body iron stores, and an increased risk of developing type 2 diabetes [78]. Nevertheless, there is limited scientific evidence indicating that the levels of these proteins used in plant-based foods pose health risks.

Concerns have also been voiced regarding the potential health risks linked to the consumption of PBMAs that incorporate a mixture of numerous additives, encompassing flavorings, colorings, binding agents, preservatives, and sweeteners. Nevertheless, there is scant evidence to suggest that this poses a significant health concern. Further research on the nutritional quality and safety of PBMAs is clearly required [67].

Certain PBMAs exhibit higher levels of saturated fat compared to conventional meat products, as well as other plant-based protein sources, like beans and lentils [18,24,54]. Indeed, some researchers argue that PBMAs exceed the recommended dietary intake for saturated fat [54] and contain approximately the same number of calories and saturated fat as livestock meat [21]. As a result, there are concerns about the potential health implications of elevating saturated fats intake in the diet. Nevertheless, it is worth noting that plant-based meat products can be reformulated to lower their saturated fats contain [1]. Moreover, there is currently ongoing debate among nutrition scientists regarding the adverse nutritional effects of saturated fats [32,41,79].

Some PBMAs contain notable elevated salt levels, which could raise health concern. Increased dietary salt levels may potentially elevate the risk of conditions, such as high blood pressure, cardiovascular disease, osteoporosis, kidney disease, and stomach cancer [14,17,80]. A high sodium content is viewed as nutritionally undesirable and, over time, may potentially increase the long-term risk of cardiovascular problems for individuals with prolonged overconsumption [81]. It may, therefore, be crucial for manufacturers to consider lowering the salt levels in their PBMA products.

PBMAs frequently feature an extensive list of ingredients [26] encompassing isolated macronutrients (proteins, fats, and carbohydrates), micronutrients (vitamins and minerals), flavoring agents, colors, emulsifiers, salt, and plant-based extracts [50,82]. The potential health effects and impact of many of these ingredients and their combinations in PBMAs remain largely unknown and uncertain.

The nutritional profile of PBMAs depends on the ingredients used by the manufacturer and can vary considerably between products. Plant proteins are often considered to have a lower nutritional quality than animal ones because of their lack of some essential amino acids and lower digestibility [5]. In addition, plant-based foods often contain lower levels of key micronutrients (e.g., iron and vitamin $B_{12}$) than the animal-based foods they are designed to replace [83]. PBMAs are lower in calcium, potassium, magnesium, zinc, and vitamin $B_{12}$ [31,54,56]. Moreover, many plant-based meat alternatives are higher in sodium and saturated fat compared to meat [24,41,54]. Consequently, there may be some nutritional concerns from switching from an omnivore to a plant-based diet. But, these concerns can often be addressed with appropriate nutritional fortification of plant-based foods, which is being carried out by some of the major producers of these products.

### 3.4. Food Safety and Nutritional Concerns

Several food safety issues can be linked with this stage of PBMA creation, including the presence of allergens, bacteria, mycotoxins, anti-nutrients, thermally induced carcinogens, and natural toxins, which are discussed in detail in the following sections.

### 3.4.1. Allergens

The presence of allergens in PBMAs is a major food safety concern [47,59]. This is particularly important due to the worldwide rise of food-related allergies over the past few decades [84–86]. Development of food allergies is believed to be based on individual reactions to food, rather than being inherited. Both food allergies and adverse reactions to food with life-threatening consequences can arise at any age and may disappear or stay throughout a person's life. Allergenic assessment of new foods is, therefore, critical to

ensure they do not pose any risks [16]. Even a small amount of certain food ingredients can cause symptoms that range from minor (such as itching, swelling, and stomachache) to severe (such as anaphylaxis) [87,88].

Common components of PBMAs can cause allergic reactions in some people [44]. For instance, many plant proteins are known allergens, including legumes and cereals, such as soy, pea, wheat, rye, barley, and lupin proteins [47,49,55,59,60,89]. The severity of the food-induced allergic reactions is impacted by the dose of the product used [47]. Some consumers are concerned that certain kinds of plant proteins obtained from genetically modified (GM) sources may introduce new allergy risks [90]. Consequently, these new proteins should undergo rigorous assessment for potential allergenicity before the foods are made available for widespread consumption [91]. The likelihood of prompting a protein-related allergic reaction is related to its resistance to digestion by the proteases in the gastrointestinal tract [91]. For transparency and avoiding consumers concerns, a clear labeling of such genetically modified food ingredients used in a product formulation is necessary [92]. Nonetheless, the health and environmental risks linked to the consumption of genetically modified foods still remain a contentious subject of debate, necessitating further research [93].

Potentially, there is also the possibility that heightened consumption of products containing substantial amounts of soy, pea, wheat, and other plant proteins could provoke and trigger allergic reactions in individuals who have not previously experienced issues with these foods [47,59,60]. High-protein pea ingredients, such as concentrated pea protein, are being formulated into PBMAs and could increase the risk of allergies [94]. Peas, being a legume, also can cause allergenicity like peanuts and lentils [95,96]. The ongoing current trend of incorporating pea protein concentrates and pea protein isolates into various foods to add bulk and increase protein content levels could potentially lead to consumption-induced allergic reactions [97,98]. Reports from individuals with peanut allergies have indicated and reported post-consumption allergic symptoms, implying that the similarity between pea and peanut proteins might trigger cross-reactions [99]. With its increased exposure, peas can develop into a more frequently encountered allergenic food [47,59,94].

Wheat proteins used in PBMAs are also a common allergen capable of inducing life-threatening severe anaphylaxis reactions [100–102], as well as less serious reactions, but with still undesirable symptoms [84]. These proteins often serve as binders in various PBMA products. Consumers with wheat allergies and celiac disease may also have an adverse allergic reaction to gluten, a protein found in grains, such as wheat, barley, and rye. Consequently, it is important to select proteins that have a low allergenicity when formulating plant-based foods and to ensure careful processing and labeling. The Codex Alimentarius Commission includes a priority allergen list within its General Standards for the Labeling of Prepackaged foods. This list is developed and formulated based on predetermined criteria, which take into account global allergen prevalence and other established factors [103,104].

### 3.4.2. Bacteria

Throughout history, bacteria have been responsible for a disproportionate share of human diseases and fatalities. Understanding the various types of bacteria linked to food is of paramount importance for ensuring food safety. Pathogenic bacteria originating from the raw ingredients can be present in PBMA products [44]. Bacteria are usually inactivated during food processing operations (such as extrusion and cooking); however, studies have shown that some endospore-forming bacteria, (e.g., *Clostridium* spp. or *Bacillus* spp.) and other bacteria (e.g., *Lactobacillus sakei* and *Enterococcus faecium*) can live through the heating regime [105] or can be present in the final product due to post-extrusion process re-contamination [44,48,51].

Bacteria not only can encompass infectious pathogens, but also toxin-producing strains, which similarly pose significant safety concerns within the field of food microbiology.

### 3.4.3. Toxins

Substances, whether they are of natural or artificial origin, can pose risks when the level of exposure reaches a certain threshold.

### Mycotoxins

Mycotoxins are hazardous toxic compounds produced by a variety of fungal (mold) species [106]. Among food commodities, the predominant mycotoxins include aflatoxins and ochratoxins, produced by Aspergillus species, as well as ochratoxins and patulin, synthesized by Penicillium. Additionally, Fusarium species generate fumonisins, deoxynivalenol, and zearalenone. Globally, mycotoxins, such as fumonisins, patulin, aflatoxins, and ochratoxins, among others, are accountable for a multitude of acute and chronic human illnesses [107].

Many edible plants are susceptible to contamination with mycotoxins, which are harmful to humans when ingested in sufficient quantities [45,108]. Indeed, many important agricultural crops have been reported to be contaminated with mycotoxins [103,109]. They are present in some of the raw ingredients used to formulate PBMAs, such as legumes (soy), cereals (oat, rice), and nuts (almond, walnut) [110]. Mycotoxins, being "mutagenic, teratogenic, and carcinogenic", are potent toxins with harmful health effects in people [107]. The severity of the toxicity is based on the exposure time, mycotoxin amount, and consumers' sensitivity [46,110].

Similar to bacteria, mycotoxins are heat resistant within the range of conventional food-processing temperatures and cannot be completely destroyed [110,111]. Ochratoxin A has been detected in plant-based foods and ingredients [112], which highlights the potential for mycotoxin contamination of PBMAs [45,46]. Another mycotoxin, Fumonisin FB1, is predominantly found in soybeans, corn, rice, beer, sorghum, cowpea seeds, and beans [110,113]. A proper regulatory framework for mycotoxins in PBMAs is a necessary step for minimizing mycotoxins contaminations and adverse health effects, such as development of life-threatening illnesses (e.g., liver cancer) [45].

### Natural Toxins

Several ingredients used in the creation and formulation of plant-based foods may harbor natural toxins. These substances are typically metabolic byproducts produced by plants as a defense mechanism against various threats, such as bacteria, fungi, insects, and predators. Common examples of these natural toxins in plants encompass lectins found in green, red, and white kidney beans; cyanogenic glycosides present in bitter apricot seeds, bamboo shoots, cassava, and flaxseeds; glycoalkaloids within potatoes; 4'-methoxypyridoxine derived from ginkgo seeds; colchicine in fresh lily flowers; and muscarine found in some wild mushrooms [114]. As a result, it becomes imperative that all plant-derived ingredients employed in product formulation undergo meticulous selection and processing procedures to prevent, eliminate, or deactivate and neutralize these toxins.

Some scholars have raised awareness regarding the utilization of carrageenan. Carrageenan is sourced from seaweed and used as an ingredient component in food products. Carrageenan, a polysaccharide, is occasionally employed in PBMAs to serve as a thickener, gelling agent, stabilizer, or binder [115]. Although food-grade carrageenan is considered safe for consumption, there have been suggestions that carrageenan might contribute to gastrointestinal inflammation, disruptions in intestinal microflora, and irritable bowel syndrome, as well as the development of colon cancer and various other health issues [116,117]. Additionally, it is possible for carrageenan to accumulate elevated concentrations of heavy metals when obtained from contaminated or polluted seawater, potentially presenting a health hazard [117,118]. Nevertheless, the existing scientific consensus about the potential health risks associated with carrageenan remains limited [50,119,120]. Despite this, the consumer desire for cleaner labels on their food products is promoting the PBMAs industry to reduce the number of ingredients of concern in their products [121]. In addition, updated

natural toxins regulations and risk assessments are needed for all PBMAs present and future products to minimize food safety risks [45].

Synthetic Toxins

Some of the ingredients used to formulate PBMAs may contain synthetic toxins, such as pesticide residues [44]. Employing organic solvents, like hexane, during the process of protein extraction can lead to both environmental and health issues, especially if relatively substantial residual amounts persist in the end product [122]. Concerns about the chemical hexane use in processing soy protein isolates are raised due to its neurotoxin nature [52]. Nonetheless, there is currently a lack of precise data available regarding the quantities of hexane used in the production of soy and pea protein isolates for plant-based alternatives, as well as the residual amounts that may persist in the final product [123]. This underscores the importance of conducting further research in this particular area.

### 3.4.4. Thermally Induced Carcinogens

Thermal processing used to reduce microbial contamination or cook foods may induce the formation of carcinogens in cooked PBMAs, just as they do in real meat products, [14]. Currently, there is little information about the formation and effects of thermally induced substances and hazardous chemicals in PBMAs [52]. Clearly, more studies are needed to verify the likely safety risk of the chemicals produced by high-temperature processing in plant-based meats [14,44].

### 3.4.5. Antinutrients

Some plant-based ingredients, such as legumes, contain antinutrients that can adversely affect human health, such as protease inhibitors, phytic acid, lectins, oxalates, goitrogens, saponins, phytoestrogens, phytates, and tannins [124]. These antinutrients can reduce the bioavailability of key nutrients by restricting protein digestion or mineral absorption [68,124,125]. Antinutrients naturally occur in some plants and may not be fully removed or deactivated during extraction and extrusion processes [125,126]. For instance, studies have shown that some may be present in maize, soybean, and cassava starch ingredients [127]. Lectins are of concern in the production of soybean and rice milk substitutes [67], but may also be a concern for the production of PBMAs. Some researchers have identified factors in plant-based proteins that may decrease nutrient bioavailability that include structures that are resistant to proteolysis, certain protein conformations, and the presence of antinutrients [53]. Therefore, testing for anti-nutrients is an important aspect when developing plant-based proteins [128].

### 3.5. Nutritional Profile

There is a potential impact on human health and nutrition from switching from animal-based to plant-based foods related to differences in their compositions, structures, and processing. Many of the studies included in this review have reported that PBMAs are highly processed foods that contain a longer list of ingredients than the equivalent animal-based products [32,36,38,58], which could lead to different nutritional outcomes.

As mentioned earlier, the composition of PBMAs can vary considerably between products, which means their health effects may be different. There may also be differences in the digestibility and absorption of PBMA and meat products [16,34]. A variety of processing operations are used to create PBMAs, which determine the composition and digestibility of the final product, including dehulling, soaking, blanching, pH adjustment, enzyme treatments, shearing, thermal processing, and size reduction [129]. Because they are highly processed materials, food producers can control the nutritional profile and digestibility of PBMAs, thereby modulating their health effects [34]. For instance, their macronutrient and micronutrient levels can be controlled, and they can be fortified with other health promoting ingredients.

The nutritional profile of PBMAs could be improved by reducing their salt content [17,18,39,49,54,130], as high sodium intake is associated with an increased risk of cardiovascular diseases [131,132]. In general, however, the amount of salt present is likely to depend on the particular type of plant-based food being considered and their producers' product formulation.

PBMAs may contain soy isoflavones compounds, such as phytoestrogens, which have been linked to some health concerns [133]. Generally, estrogens are considered to have beneficial effects in preventing cardiovascular disease, osteoporosis, breast cancer, and menopausal symptoms. However, it is unclear whether ingested phytoestrogens behave like endogenous estrogens in the human body [133]. Some researchers reported that when phytoestrogens are excessively consumed, they may provoke adverse health effects on reproductive health [67].

Plant proteins possess distinct amino acid profiles compared to meat proteins, potentially influencing their nutritional implications. Some of them may be deficient in one or more essential amino acids vital for human health, which the human body cannot synthesize on its own, such as methionine, lysine, and tryptophan) [72]. In principle, this deficiency could raise health concerns, but the majority of vegan or vegetarian diets incorporate a wide variety of a diverse range of protein sources, generally providing adequate levels of these essential amino acids, rendering this matter largely non-consequential.

PBMA products contain different carbohydrate ingredients from starches, flours, and binding agents. Although starchy foods are an essential part of a nutritious diet, providing energy and fiber, they can be detrimental for human health, e.g., for people with medical conditions such as diabetes [134]. Also, when cooked at high temperatures (e.g., frying, roasting, and baking), starchy, plant-based foods produce some potentially harmful chemicals, such as acrylamide [135], which could be the case with cooked PBMAs. Limiting consumer exposure to acrylamide could be achieved by avoiding high-temperature cooking and practicing storing foods in a cool, dry place.

Meatless products can be designed and formulated using liquid smoke flavorings, which have been documented to be associated with potential carcinogenicity when consumed regularly at sufficient quantities [130]. Furthermore, plant-based meat alternatives have also been observed to be deficient in certain specific amino acids and their derivatives, including creatine, taurine, and anserine. These compounds are believed to play an important, meaningful role for human health, as they can have an impact on both brain and muscle functions [19].

Food manufacturers used various additives to improve the look, feel, and taste of foods. Among these are binding ingredients or gums, which serve as emulsifiers, stabilizers, binders, and thickeners, such as methylcellulose, acacia gum, xanthan gum, carrageenan, and others. The health and safety of these products have been challenged, and consumers are demanding cleaner labels [121]. Despite no real health risks or concerns having been detected [50], food processors are already trying to limit or eliminate the use of these ingredients in many processed food products [121].

Enhancing the nutritional profiles of PBMAs can be achieved through fortification with specific nutritional components as desired. Certain essential minerals, like iron, zinc, magnesium, and calcium, may exhibit reduced bioavailability in some of the ingredients present in these alternatives [136,137]. Consequently, it is important to develop strategies to increase the bioavailability [41], as well as to have in vitro and in vivo methods to measure the bioavailability of minerals and other nutrients in PBMAs.

Many kinds of natural colorants (e.g., leghemoglobin, red beet, red cabbage, etc.) and flavorings (e.g., herbs and spices) are used in PBMAs to reproduce the desirable color and flavor of real meat [14]. These ingredients are often less stable than synthetic alternatives and may chemically degrade during food processing, leading to unacceptable changes in quality attributes [28,82]. At present, there has been little research on the potential safety aspects of the chemical degradation of natural colorants or flavorings in plant-based foods.

*3.6. Processing*

Processing influences changes of the nutritional, physical, and chemical properties of foods [50]. Many technological and food engineering approaches exist to create the plant proteins texture, but balancing the processing methods to achieve all the desired mechanical properties and, at the same time, to retain the final product nutritional value still remains difficult [68]. Ishaq and colleagues highlighted that contemporary methods of structuring have substantially enhanced the operational capabilities of plant-based meat substitutes. Nonetheless, significant efforts are still required to enhance their operational efficiency, sensory qualities, safety, and the choice of appropriate components [28].

Typically, plant proteins, commonly in a defatted state, are combined with water, carbohydrates, salts, flavorings, and edible fats. This mixture is then subjected to a twin-screw extruder operating at high temperatures and varying moisture levels. This process encourages the proteins to form a meat-like fibrous structure, resulting in the creation of a meat substitute suitable for various food applications [55]. PBMAs processing concerns are discussed in more detail.

According to the description of ultra-processed, PBMA products fit entirely into the portrayal, as they are "formulations made mostly or entirely from substances derived from foods and additives, with little if any intact (whole) food" [77]. Ultra-processed foods consumption is associated with many adverse health consequences, such as obesity, cardiovascular disease, cancer, type 2 diabetes, and all-cause mortality [23]. On top of the highest degree of processing, PBMAs are inclined to contain a greater diversity of other ingredients needed to mimic the characteristics and sensory attributes of conventional animal-based products. This all provides grounds for a variety of sources, from where prospective hazards can develop and arise. The argument of the ultra-processed nature of the PBMAs and its connection to a potential elevation risk of health-related harm, affecting the consumer motivation for their regular consumption, is an imminent part of the discussion around these alternative products' pivotal place in the transition toward more sustainable dietary change [36,138,139].

Different degrees of processing have a serious impact on health and nutrition, as many nutrients, vitamins, and minerals can be destroyed or removed during the process. Greater understanding of the possible impact is necessary, as even adding some specific ingredients designed to enhance nutritional quality (e.g., fortification) can lead to reduced product desirability among consumers [13]. Processing can increase or decrease the bioavailability, digestibility, nutritional, and functional characteristics of particular foods and ingredients [62,128]. To this should be added the long ingredients list and the tolerance for discrepancies from the food label laws worldwide regarding the actual nutritional value and the values described in the PBMAs food labels [39].

Like meat products, it is also important to subject plant-based meat analogues to sufficient and adequate thermal processing prior to consumption to ensure and guarantee their microbiological safety [43]. This thermal treatment can be conducted in a factory, restaurant, or even at home. The potential for microbial proliferation and presence of yeast and Enterobacteriaceae after hot meals preparation with PBMAs also needs special attention, as some researchers observed it to be slightly higher than that of meat-containing food [43]. For consumers, adhering to the food preparation instructions provided by the manufacturer on food labels is important because legumes, grains, and vegetables have the potential to be contaminated with pathogenic bacteria [48]. Observing sound food safety practices ensures that these foods pose no harm to consumers when prepared and consumed in accordance with their intended use.

Most PBMAs are processed using extrusion [140], such as dry extrusion, wet or high moisture extrusion, and thermal extrusion, or power heated [141,142]. Conventional dry extrusion is an established processing technique well suited for producing minced meat substitutes. Nonetheless, emerging high-moisture extrusion technology enables the creation of appealing whole pieces of alternative meats. Extrusion represents a high-temperature and high-pressure technique used to achieve the desired form and texture of products, while

also simultaneously decreasing the microbial load [143]. The precise time and temperature factors involved are considered and regarded as critical control points. Additional research investigation into non-protein constituents, advancements, and innovations in production technologies for alternative protein products, improvements, and enhancements in overall appearance and flavor, rigorous control over biological and chemical safety, and the careful selection of protein sources are all essential for addressing food safety and quality issues and for the ongoing continuous expansion and diversification of protein offerings in the marketplace.

There are several chemical hazards that could potentially arise from the processing of PBMAs. Known carcinogens, such as the heterocyclic aromatic amines, nitrosamines, and polycyclic aromatic hydrocarbons, could be produced during thermal processing of PBMAs, just as they are produced in real meat products [14]. Similarly, other heat-induced contaminants could be produced in PMBAs during thermal processing, such as glycidyl fatty acid esters, 2-monochloropropanediol (2-MCPD), and 3-monochloropropanediol (3-MCPD) [144,145]. However, the production and effects of these kinds of toxic compounds in PBMA requires further investigation.

The utilization of lipid sources containing trans-fatty acids, which are formed during partial hydrogenation of vegetable oils, may also adversely affect the healthiness of PMBAs [146]. However, many countries around the world have legislation in place to ban industrially produced artificial trans-fatty acids from their food products [147]. Food processing may lead to the loss of certain beneficial nutrients and phytochemicals in plant-based foods, which could reduce their potential health benefits. In general, van Vliet et al. [19] advises against classifying plant-based alternatives as nutritionally equivalent to their corresponding animal-based counterparts [37]. Their metabolomics study indicates that the animal-based product (beef) and the PBMAs are more likely to be complementary, rather than identical or interchangeable, in terms of providing beneficial nutrients.

Another issue of concern is the digestion and bioavailability of PBMA products [41,148]. Some researchers believe that the industrial processing of plant-derived ingredients to form meat alternatives may not necessarily be unfavorable, as it has the potential to promote and encourage favorable alterations and positive changes in protein digestibility, nutrient bioavailability, and the human gut microbiome [25,41]. Others believe that the novel protein sources can potentially trigger adverse allergen and other reactions and, therefore, incontestably require thorough risk assessment [148], as well as an overall multidisciplinary approach [149].

### 3.7. Storage

The influence of microbial contamination on the safety of PBMAs is another issue of concern. Like meat products, PBMA products should be stored under appropriate conditions (e.g., temperature, humidity, packaging) to reduce the growth of undesirable microbial contaminants [105]. PBMAs are generally not strongly associated with concerns related to pathogenic diseases. However, they can potentially cause foodborne illnesses. For instance, they may become contaminated with pathogens through contact with sources like animal manure, water, or other foods [53]. PBMAs typically have a neutral pH, high moisture content, and a favorable nutrient profile, which makes them highly susceptible to microbial growth and spoilage [51,105].

At present, there is currently a lack of research on the microbial contamination and safety of plant-based foods, and more research is clearly needed [92,150]. Researchers have employed meta-genetics to examine shifts in the quantities and types of microorganisms in plant-based products during storage [151]. Some studies have noted the increased prevalence of particular microbes towards the end of the product's shelf-life period, including *Latilactobacillus sakei*, *Enterococcus faecium* [152], and *Enterobacteriaceae* and yeasts [153]. These microbes may be present as a result of post-processing contamination of heat-treated PBMAs. Improved knowledge and understanding of the types of microorganisms present is crucial for ensuring the safety of PBMAs.

## 4. Studies Comparison

In the analysis of 48 studies concerning the safety and nutritional risks of plant-based meat alternatives, three distinct categories of research were identified, each contributing to evidence-based insights on these aspects. The majority of studies within the reviewed literature (27 in total) primarily concentrated on nutritional risks. Additionally, there were 15 articles that provided comprehensive information and analysis covering both nutritional and safety concerns. However, it is worth noting that a relatively small number of studies (six in total) specifically delved into the safety risks associated with plant-based meat alternatives.

In the assessment of these issues, a noteworthy number of consensuses and only a few points of controversy emerged, particularly within the studies that centered on nutritional risks (27 in total). The areas of agreement were predominantly related to several key concerns regarding plant-based meat alternatives (PBMAs). These included the higher levels of sodium, carbohydrates, and saturated fat content found in PBMAs, as well as the deficiency of essential nutrients, vitamins, and minerals in these products. This consensus was supported by numerous studies [17–19,21,22,24,26,28,34,36,41].

Another commonly shared concern revolved around the quality of protein in PBMAs and the necessity for optimizing the nutritional profiles and functional properties of these alternatives [29–31]. Additionally, there was a commonly acknowledged consensus regarding the importance of product information disclosure by manufacturers, particularly in relation to potential health aspects associated with PBMAs [21]. This included transparency in claims around the GMO-free claims [23].

Among the studies identified, there was a lack of consensus regarding the classification of plant-based meat alternatives (PBMAs) as ultra-processed products. Some publications argued against categorizing all PBMAs as detrimental to the human gut microbiome [25], suggesting that their consumption as part of a flexitarian diet could potentially promote positive changes in the microbiome. Conversely, other scholars contended that the higher degree of processing in PBMAs raises concerns about their long-term health implications [27,32,36–38,41], including organically certified PBMAs [42]. Additionally, the extensive list of ingredients in PBMAs was cited as a factor contributing to their classification as ultra-processed products [33,37].

The composition of ingredients in PBMAs and the role of excessive fibers also emerged as contentious topics within the discourse on plant-based meat alternatives. According to some researchers, an overabundance of fibers could negatively impact their taste and, consequently, their marketability and commercialization [39].

Among the studies that primarily focused on safety risks (six in total), the research was primarily centered on several critical aspects, including microbial proliferation, the formation of pathogenic bacteria, mycotoxins, chemical contaminants, and natural toxins [43–46,48]. It appeared that all of these studies were aligned in emphasizing the necessity for a comprehensive regulatory framework to address safety risks associated with plant-based meat alternatives (PBMAs), particularly in relation to mycotoxins [45,46], due to concerns about potentially severe health risks.

Another noteworthy and shared area of concern within these safety-focused studies revolved around the potential creation of PBMAs with allergenic potential [47]. This raised critical questions regarding the allergenicity of certain ingredients used in PBMA production and its implications for consumer safety.

In the studies that explored both nutritional and safety risks associated with PBMAs (*n* = 15), the authors echoed the concerns previously mentioned. Additionally, these mixed studies introduced a new perspective, emphasizing the need for further research to understand the implications of variations in protein digestion, absorption, amino acid profiles, and allergenicity on human health and well-being. This highlights the evolving nature of our understanding of the potential impacts of PBMAs on various aspects of human health, underlining the importance of continued investigation in this field.

## 5. Conclusions

Currently, there is insufficient information about the potential safety and nutritional health impacts of consuming plant-based meat alternatives with different nutritional profiles to animal-based ones, and further research is clearly needed. In addition, there are new safety concerns associated with incorporating next-generation plant-based foods into the human diet, which arise from the potential presence of allergens, chemical contaminants, antinutritional factors, and pathogenic microorganisms. The long-term health effects of regular PBMA consumption are also not yet comprehensively assessed, and further investigation is clearly required.

More research is needed to identify potential safety and health concerns associated with the new generation of PBMAs, as well as to develop technological innovations to mitigate any potentially adverse effects. Moreover, there is a need to establish appropriate food standards and guidelines, and to create adequate risk assessment and management methods. This is particularly relevant for emerging foods, where there is a limited understanding and knowledge of their risks and advantages. This lack of knowledge hampers harmonizing regulatory frameworks to address safety concerns and to guide the safe application of these products. Progress in this area will be made only if an integrated multidisciplinary approach is considered to help overcome the various challenges and enable the responsible advancement of next-generation plant-based meat alternatives.

In the future, it will be important to specifically design PBMA products to improve human health and well-being, as well as for being delicious and sustainable, which will require the development of new product formulations and processing methods. Further research on nutrient bioavailability, safety, costs, and consumer acceptance will shape the future of plant-based foods in future human diets.

**Author Contributions:** Conceptualization, D.B. and D.J.M.; methodology, D.B.; formal analysis, D.B.; investigation, D.B. and D.J.M.; writing—original draft preparation, D.B.; writing—review and editing, D.J.M. All authors have read and agreed to the published version of the manuscript.

**Funding:** This research received no external funding.

**Institutional Review Board Statement:** Not applicable.

**Data Availability Statement:** Data included within the text.

**Conflicts of Interest:** The authors declare no conflict of interest.

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
