# Peer review of "Safety and Nutritional Risks Associated with Plant-Based Meat Alternatives"

_sustainability, doi:10.3390/su151914336_

Round 1

Reviewer 1 Report

In this review the authors addressed the question of food safety and health risks for a rather new group of products – plant-based meat alternatives (PBMA). They reviewed the existing literature and make analysis and recommendations in regard to these products. This has been done by analyzing the food safety issues (chemical, microbiological, toxins, allergen issues) and nutritional aspects of these products. In addition, the authors nicely described the process of production of these products and analyzed and discussed potential food safety and nutritional issues for each phase of production.

This gave a rather big and clear picture about food safety and nutritional characteristics and issues for PBMA products.

The topic is relevant to the field because it gives a wide, clear and comprehensive review of food safety and nutritional issues for PBMA products with recommendations where to pay attention in the production of these products. Because most of research is focused on preparation, formulation and production of these products there are not many reports on their safety and nutritional issues like in this review. The review adds to the current knowledge and gives a clear picture on these issues.

The authors used adequate body of literature to review and analyze this topic and to give opinions and recommendations.

The chosen methodology is well selected, described in detail and well presented.  

The manuscript is well organized, logical and systematic. Table and figure are well done and give additional quality and fit well the manuscript.

Final conclusions are appropriate and based on the material described in the review and they are in accordance with topic of the review.

References used in the review are relevant and adequate. 

Author Response

We thank the reviewer for this positive response and comments. 

Reviewer 2 Report

The abstract should clearly provide the main conclusions of the paper.

Please change Table 1 to a textual description. The current form is not a scientific method of expression.

Simple references presentation cannot produce high-quality reviews. As shown in Table 2 (there are two Tables 1 in the manuscript. Error). The author needs to classify and organize the references, provide an overview of the progress, deficiency, and possible research direction in each branch. Based on this, propose your own unique insights.

Table 2 is too cumbersome, with up to 13 pages. Does not meet general paper standards. It is recommended to include it in the attachment.

Change Figure 1 to a textual description.

The first paragraph of the Results and Discussion section is suggested to be rewritten. It does not serve to connect the context.

Please provide standardized numbering for titles at different levels, such as lines 136-150.

Titles at all levels should use declarative sentences or noun phrases.

Some descriptions are too cumbersome or repetitive. It is recommended to simplify or delete them, such as lines 108 to 134.

For the introduction of a certain branch, it is recommended to first provide an overall overview and then gradually expand in a certain order or logic. Instead of simply listing the literature. As in 3.3

The shortcomings mentioned earlier lie in almost every part of the paper. Suggest making comprehensive revisions to the paper. I don't have time to point out the relevant issues one by one.

The conclusion needs to be greatly simplified and the key points highlighted. Furthermore, references should not appear in this section.

The format of the references is exceptionally chaotic, which is an error that should never occur in a review paper.

Author Response

The abstract should clearly provide the main conclusions of the paper. - As requested, we have revised the abstract to include the conclusions of the paper.

Please change Table 1 to a textual description. The current form is not a scientific method of expression. - As requested by the reviewer, we have removed Table 1 from the revised manuscript and included the information as a description in the text.

Simple references presentation cannot produce high-quality reviews. As shown in Table 2 (there are two Tables 1 in the manuscript. Error). The author needs to classify and organize the references, provide an overview of the progress, deficiency, and possible research direction in each branch. Based on this, propose your own unique insights.

As suggested, we have revised the discussion selection of the revised manuscript to provide more discussion and critical assessment of the articles cited.

Table 2 is too cumbersome, with up to 13 pages. Does not meet general paper standards. It is recommended to include it in the attachment. - In response, the length of the table presented has been reduced by separating it into three types of studies. Now there is a table only with nutritional risks which is only 2.5 pages, another table with safety risks is 1 page and a third table is for study discussing both risks and now is 4.5 pages.

Change Figure 1 to a textual description. - In response, a textual description of the Prisma chart has been included in the revised manuscript. The following text is added: “Figure 1 uses a PRISMA flow chart to document the selection criteria used in this study. The chart maps out the number of recorded studies and visually illustrates the flow of the studies selections though each phase of the review process in sequential order. From a total number of n=327 articles initially identified, the final list of articles and topics related to safety and nutritional risks were reduced to n=48 articles. These are discussed in association with the main PBMAs production stages such as: (1) protein isolation and functionalization; (2) product formulation; (3) processing, and (4) storage”.

The first paragraph of the Results and Discussion section is suggested to be rewritten. It does not serve to connect the context. - As requested, this section has been rewritten.

Please provide standardized numbering for titles at different levels, such as lines 136-150. - As requested, we have used standardized numbering throughout the revised manuscript.

Titles at all levels should use declarative sentences or noun phrases. - In response, we have revised the title of the sections.

Some descriptions are too cumbersome or repetitive. It is recommended to simplify or delete them, such as lines 108 to 134. - As requested, we have revised these descriptions to make them more clear and concise.

For the introduction of a certain branch, it is recommended to first provide an overall overview and then gradually expand in a certain order or logic. Instead of simply listing the literature. As in 3.3  We have revised the manuscript as suggested by the reviewer. The following text is added: Product formulation entails understanding how materials interact to produce enhanced properties, optimize processing efficiency, and deliver active ingredients effectively. In the context of food, formulation pertains to the art of crafting, planning, or evolving food items with the aim of incorporating specific functionalities. These functionalities can span from conferring extra nutritional advantages to enriching food products. PBMAs are distinctly defined as products meticulously crafted to emulate the color, flavor, taste, aroma, consistency, texture and visual characteristics of animal-derived products to match or at least closely align with the sensory encounter of consuming meat products.

The shortcomings mentioned earlier lie in almost every part of the paper. Suggest making comprehensive revisions to the paper. I don't have time to point out the relevant issues one by one. In response, we have read through the manuscript and revised it throughout. The text is reviewed and an overall overview is provided as above recommendation to all the branch where they were missing.

The conclusion needs to be greatly simplified and the key points highlighted. Furthermore, references should not appear in this section. We have made the change requested by the reviewer.

The format of the references is exceptionally chaotic, which is an error that should never occur in a review paper. Our apologies, the references format has been revised as suggested.

Reviewer 3 Report

Good work but a few specific comments are as follows:

Page 2, Line 58…………write reference according to journal guideline…….(Marinova and Bogueva, 2022 [6])………..correct to ………..[6].

Page 3, Lines 89-91………………. Is the total study sample n=48? How the author categorized them into “n=42 and n=21The selected articles focused on both the nutritional risks (n=42) and the safety risks (n=21) of PBMAs……..Are there articles that share both nutrition risks and safety? Please clarify this point

Also the table: Table 1. Summary characteristics of the included articles related to PBMAs’ safety and nutrition risks aspects discussed……………number it Table 2, because there is table 1 before.

Table 1: I suggest mentioning articles of safety (First), then those related to nutritional risks only (second), and at the end of table 1 mention articles referring to both…………..this will be easy for readers.

Page 3, Table 1: non-GMO diet………..define each abbreviation for the first time.

Page 5: table 1……. 7 Luchansky et al., 2020 [21]………..Comment pathogen viability ………….correct……….. Common pathogen viability

Page 24, Lines from 476- needs a reference or more.  

Best 

Author Response

Page 2, Line 58…………write reference according to journal guideline…….(Marinova and Bogueva, 2022 [6])………..correct to ………..[6]. Corrected

Page 3, Lines 89-91………………. Is the total study sample n=48? How the author categorized them into “n=42 and n=21”The selected articles focused on both the nutritional risks (n=42) and the safety risks (n=21) of PBMAs……..Are there articles that share both nutrition risks and safety? Please clarify this point  Thank you for this. That is correct. There are articles that share both nutrition and safety risks. We added an additional text to L91 “…with some of them sharing information and analysis on both risks”.

Also the table: Table 1. Summary characteristics of the included articles related to PBMAs’ safety and nutrition risks aspects discussed……………number it Table 2, because there is table 1 before. Thanks for noting. It is corrected now.

Table 1: I suggest mentioning articles of safety (First), then those related to nutritional risks only (second), and at the end of table 1 mention articles referring to both…………..this will be easy for readers. Good suggestion. Now there are three tables one for nutritional risks, one for safety risks and the third table for studies discussing both risks.

Page 3, Table 1: non-GMO diet………..define each abbreviation for the first time. “genetically modified organism (GMO)” is added

Page 5: table 1……. 7 Luchansky et al., 2020 [21]………..Comment pathogen viability ………….correct……….. Common pathogen viability Thanks you for noting. Now it is corrected.

Page 24, Lines from 476- needs a reference or more.  References are added:
Ozturk OK, Hamaker BR. Texturization of plant protein-based meat alternatives: Processing, base proteins, and other constructional ingredients, Future Foods, Volume 8, 2023, 100248,
https://doi.org/10.1016/j.fufo.2023.100248. https://www.sciencedirect.com/science/article/pii/S2666833523000345

Arora S, Kataria P, Nautiyal M, Tuteja I, Sharma V, Ahmad F, Haque S, Shahwan M, Capanoglu E, Vashishth R, Gupta AK. Comprehensive Review on the Role of Plant Protein As a Possible Meat Analogue: Framing the Future of Meat. ACS Omega. 2023 Jun 20;8(26):23305-23319. doi: 10.1021/acsomega.3c01373. 

Schmid EM,  Farahnaky A,  Adhikari B., Torley PJ. High moisture extrusion cooking of meat analogs: A review of mechanisms of protein texturization Comprehensive review in food science and food safety, 19 September 2022.  https://doi.org/10.1111/1541-4337.13030

Verma T, Subbiah J. Conical twin-screw extrusion is an effective inactivation process for Salmonella in low-moisture foods at temperatures above 65 °C, LWT, Volume 114, 2019, 108369, https://doi.org/10.1016/j.lwt.2019.108369.

Author Response

  1. Introduction section need to improve and add some more references of studies publishes in tire 1 Journals Improved and more references added across the manuscript.
  1. Add a comprehensive table and a figure to enrich your

Not clear what the exact recommendation is but a separation between the three types of studies selected was done. Now there is a table only with nutritional risks, another table with safety risks and a third table is for study discussing both risks. Thank you for the suggestion.

  1. The conclusion section need to reduce which only represent main findings The conclusion section is reduced as suggested

  2. Line 106 change heading .1. How are plant-based meats made?. There should not question in heading This is corrected now.
  1. Comparison of different studies should be included in result and discussion section As required a news section 4. Studies comparison was added.
  2. The whole article need to check for grammar and typos error

This is done as suggested.

  1. All references must be according to journal guidelines The references are reviewed and fixed as suggested.

Round 2

Reviewer 2 Report

The author has made in-depth revisions to the manuscript, resulting in a significant improvement in the quality of the paper. The proportion of tables is still relatively large.

Author Response

Thank you for the comment. The text volume in the tables was reduced to shorten the existing tables.